# Suicidal Offenders and Non-Offenders with Schizophrenia Spectrum Disorders: A Retrospective Evaluation of Distinguishing Factors Using Machine Learning

**DOI:** 10.3390/brainsci13010097

**Published:** 2023-01-04

**Authors:** Lena Machetanz, Steffen Lau, Elmar Habermeyer, Johannes Kirchebner

**Affiliations:** Department of Forensic Psychiatry, University Hospital of Psychiatry Zurich, 8008 Zurich, Switzerland

**Keywords:** machine learning, advanced statistics, schizophrenia spectrum disorders, offender patients, forensic psychiatry, suicidality, suicide

## Abstract

Patients with schizophrenia spectrum disorders (SSD) have an elevated risk of suicidality. The same has been found for people within the penitentiary system, suggesting a cumulative effect for offender patients suffering from SSD. While there appear to be overlapping characteristics, there is little research on factors distinguishing between offenders and non-offenders with SSD regarding suicidality. Our study therefore aimed at evaluating distinguishing such factors through the application of supervised machine learning (ML) algorithms on a dataset of 232 offenders and 167 non-offender patients with SSD and history of suicidality. With an AUC of 0.81, Naïve Bayes outperformed all other ML algorithms. The following factors emerged as most powerful in their interplay in distinguishing between offender and non-offender patients with a history of suicidality: Prior outpatient psychiatric treatment, regular intake of antipsychotic medication, global cognitive deficit, a prescription of antidepressants during the referenced hospitalisation and higher levels of anxiety and a lack of spontaneity and flow of conversation measured by an adapted positive and negative syndrome scale (PANSS). Interestingly, neither aggression nor overall psychopathology emerged as distinguishers between the two groups. The present findings contribute to a better understanding of suicidality in offender and non-offender patients with SSD and their differing characteristics.

## 1. Introduction

Schizophrenia spectrum disorders (SSD) are severe mental disorders with a substantial burden of disease due to significant impairments in many domains and high excess mortality, with 10 to 25 years less life expectancy than the general population [1,2]. While natural causes, such as respiratory and cardiovascular diseases, account for most deaths in this population, another risk factor for early mortality is the high rates of suicidality among patients with SSD, with a lifetime prevalence of around 30% for suicidal ideations, suicide plans and attempts, and a lifetime risk of suicide between 3–7% [3,4,5,6,7]. Factors known to contribute to the high risk of suicide for patients with SSD are young age, male gender, a high level of education, a history of suicide attempts as well as depressive and delusional symptoms [7]. Further studies suggest an association between violent behaviour and suicidal or self-harming behaviour in general among patients with SSD [8,9]. Results regarding the influence of hallucinations are mixed: while some authors have described them to contribute to an elevated risk, a systematic review by Hawton et al. found an association between a reduced risk of suicide under hallucinatory symptoms [7,10].

Another population at higher risk for suicide than the general population are people who display aggressive and violent behaviour in general, and violent offenders in particular [11,12,13]. In these populations, having a psychiatric disorder, such as depression or substance use disorders, has been found to be associated with higher rates of suicide [14]. A large cohort study among prisoners in the United States found a substantially higher suicide risk if schizophrenia (RR = 7.3) and other non-schizophrenic psychotic disorders (RR = 13.8) were present [8]. A longitudinal Danish study amongst offenders identified custodial sentencing, but, even more so, sentencing to psychiatric treatment to be strongly associated with increased suicide risk (OR = 26.65) [15]. A smaller study comparing prison and forensic psychiatric hospital populations confirmed a higher suicide rate in the latter [9]. These previous findings suggest that, among patients with SSD, the subgroup of offender patients with SSD is at particularly high risk for suicide and suicidal behaviour. Even though offender patients with SSD can be considered a dual-vulnerable population due to the coercive treatment context as well as their often highly impairing mental disorder, little is known about whether this subgroup is influenced by other mediators with respect to suicidality, and whether the known risk factors have similar weights as in general psychiatric patients. Previous literature has either focused mainly on inmates or patients with SSD, but rarely populations with both these features in co-occurrence. In a recent study, our research group evaluated predictors of self-harm in offender patients with SSD and identified the early onset of disorder and higher severity of psychopathology as risk factors [16]. As this had only been an explorative analysis within the offender population, there was no control group.

This research gap cannot be closed by simply applying knowledge gained from general psychiatric patients due to numerous systematic differences between forensic and general psychiatric patients: First of all, OP are known to have a higher rate of comorbidity, especially substance use [17,18]. Secondly, they receive treatment in a compulsory context as it is court-mandated, and the treatment goals may not be in line with the patients’ own ideas [19]. Thirdly, while aggressive behaviour is also common in acute psychiatric wards, OP stand out amongst psychiatric patients due to their history of severe violence [20,21]. They also are more likely to be subjected to disadvantageous social circumstances, such as unemployment and homelessness [20]. Due to the fact that there are several key differences between OP and NOP, it can be assumed that the two groups differ in many domains, including suicidality.

Therefore, and in light of previous research, the first objective of this study now presented was to detect the most distinguishing factors between offender patients with SSD (OP) and non-offender patients with SSD (NOP), both with a history of suicidality. Our second objective was to build a model based on these distinguishing features and to rank them according to their significance, using advanced statistics in the form of machine learning (ML). This study was reviewed and approved by the Ethics Committee Zurich [Kanton Zürich] (committee’s reference number: KEK-ZH-NR 2014–0480).

## 2. Materials and Methods

The study group comprised 370 offender patients (OP) with an SSD, including schizophrenia, schizoaffective and delusional disorders (F2x according to ICD-10 and ICD-9), who had all been in court-mandated treatment at the Centre for Inpatient Forensic Therapies of the University Hospital of Psychiatry Zurich, Switzerland due to being found not guilty by reason of insanity (NGRI) or for treatment of acute syndromes while being in the penitentiary system [22,23]. This population has been used in previous exploratory analyses of our research group as part of a larger ongoing project studying offender patients with SSD. The comparison group comprised 370 non-offender patients (NOP) with SSD, who had been in inpatient treatment at the Centre for Integrative Psychiatry of the University Hospital of Psychiatry Zurich. This general psychiatric facility focuses on rehabilitative treatment, and its population mostly consists of patients suffering from chronic and/or prolonged courses of disorder. As this is a characteristic shared with the majority of forensic psychiatric patients, we deemed this sample to be particularly suitable as a comparison group. Another reason was that in both groups, initial treatment for acute psychosis had already been established in most cases, either in an acute ward for NOP or in a prison setting for OP.

Data from the files of all patients were retrospectively assessed through directed qualitative content analysis [24]. Data extraction was performed by two experienced psychiatrists according to an adapted rating protocol based on a set of criteria originally described by Seifert and Nedopil [25]. The case files were rather comprehensive and included professionally documented medical histories, psychiatric/psychologic reports of both hospitalisations as well as outpatient treatments, extensive progress reports by clinicians, nursing and care staff, as well as—in the case of the OP population—testimonies, court proceedings and data regarding previous imprisonments and detentions. Data on the following areas were collected from said case files: social-demographic data, childhood/youth events, psychiatric history, past criminal history, social/sexual functioning, prison data, and particularities of the current hospitalisation and psychopathological symptoms defined by an adapted three-tier positive and negative syndrome scale (PANSS) [26,27]. The PANSS was developed as a multidimensional rating scale for assessing positive and negative syndromes as well as overall psychopathology in patients with schizophrenia and is validated and globally well established, with a mean interrater correlation of 0.83–0.87 and internal reliability of 0.73–0.83 (α coefficient) [27]. For this study, we used an adapted three-tier rating scale instead of rating each item on a seven-tier scale. As we wanted to match both groups by gender, no further female patients were included in the NOP sample after having reached the number of female patients in the OP group.

Suicidality was defined as either one of the following before the offence leading to the referenced forensic psychiatric hospitalisation for OP, respectively, before the referenced hospitalisation for NOP: suicidal thoughts and ideations, plans, and attempts. This information was also assessed retrospectively through the patients’ case files (e.g., reports of previous psychiatric hospitalisations). After omitting all cases without a history of suicidality, a total of 399 patients remained, with 232 of them being OP (58.1%) and 167 NOP (41.9%). In the initial step, we calculated the *p*-value for demographic and psychiatric variables to evaluate basic group characteristics. For this purpose, we performed an independent samples Mann–Whitney-U-Test for all metric variables with non-normal distribution, and a Fisher’s Exact Test for all other variables [28,29].

Predictor variables were selected in accordance with previous findings (see Appendix A for a detailed overview of our selected variables and their reference in previous literature).

We applied supervised machine learning (ML) to evaluate the interplay of the variables and to rank them according to their contribution to the model. Supervised ML learning is used generally to describe prediction tasks in order to classify a specific outcome of interest—in this case, suicidality—as opposed to unsupervised ML, which can be useful to find relationships in a dataset without having measured an outcome [30]. The performance of said model regarding its ability to differentiate between suicidal OP and NOP should then be quantified. An overview of the statistical steps is shown in Figure 1 provides a first glance at the statistical procedures step-by-step. Due to overlapping methodological approaches, part of the following section has been previously published, e. g. in Hofmann et al. [26], and is therefore partially replicated here.

All the steps were performed using R version 3.6.3. (R Project, Vienna, Austria) and the MLR package v2.171 (Bischl, Munich, Germany). The balanced accuracy was calculated in MATLAB R2019a (MATLAB and Statistics Toolbox Release 2012, The MathWorks, Inc., Natick, MA, USA). Initially, we processed all raw data for ML (see Figure 1, Step 1): Several categorical variables were converted to binary code., while continuous and ordinal variables did not undergo adjustment. The independent variable was dichotomised into (a) “offender patient (OP)” and (b) “non-offender patient (NOP)”, with the latter being defined as the positive class. All variables with missing values of 30% or more were omitted, yielding 107 possible predictor variables (for a detailed variable description, please refer to the material in the Appendix A and Appendix B). In a second step, the database was split into one training subset containing 70% of all cases, and one validation subset containing the remaining 30% (see Figure 1, Step 2). The training subset was used for variable reduction and model building/selection. To avoid the need to omit variables entirely due to missing values in an already small population, we conducted an imputation of missing values (by mean for continuous variables, and by mode for categorical variables). Imputation weights were stored to be later reapplied to the validation subset (see Figure 1, Step 3a). As we wanted to identify the most influential predictors and as a decrease in variables can counter-act overfitting while maintaining practical computing times in initial model building, we performed a variable reduction through random-Forest SRC, down to the point where the AUC did improve by no more than 5% through adding another item (see Figure 1, Step 3b). This led to a variable reduction down to the six most predictive variables. With the database of n = 399 being relatively small for ML purposes, we applied discriminative model building with seven algorithms (see Figure 1, Step 3c). The quality of each model was assessed in terms of established performance parameters. The model with the highest AUC was then chosen for final model validation with the validation subset (see Figure 1, Step 3d). Finally, with the intention to prevent overfitting, we used a nested resampling model with the inner loop performing imputation, variable filtration, and model building within fivefold cross-validation, and the outer loop for performance evaluation also embedded in fivefold cross-validation (see Figure 1, Step 4). This cross-validation artificially created five different subsets of our dataset of equal size so that one subset could serve as the training set for our model, while the remaining four subsets allowed us to evaluate the accuracy of the learned model [31,32]. To evaluate the model selected earlier, we applied it to the validation subset, which included 30% of all cases: The imputation weights from Step 3a were reused on the validation subset (see Figure 1, Step 5). Afterward, the selected model was applied (see Figure 1, Step 6). Lastly, the variables identified as most dominant in the model were finally ranked according to their indicative power (see Figure 1, Step 7).

## 3. Results

### 3.1. Demographic and Psychiatric Characteristics

As shown in Table 1, the two groups were well balanced regarding demographics: Around three-quarters of both OP and NOP were diagnosed with schizophrenia, while the remaining one-quarter were admitted under diagnoses of schizoaffective disorder, acute psychotic disorder, and other disorders from the schizophrenia spectrum (F2x. according to ICD-10 and ICD-9). Both groups were predominantly single at the time of their criminal offence (OP), respectively, of their admission to the referenced hospitalisation. NOP, however, had a lower rate of patients whose native country was Switzerland. Around half of the total population had at least one documented suicide attempt prior to their referenced hospitalisation, a rate that was also reflected in the individual groups. Suicide attempts during the referenced hospitalisation were extremely rare, with 2 in the NOP and 10 in the OP group. As expected, OP expressed higher rates of endangerment of others both in the past as well as during the referenced hospitalisation.

### 3.2. Model Calculation Using Machine Learning (ML)

The performance parameters of the seven calculated ML algorithms are provided in Table 2.

With a balanced accuracy of 76.6% and an AUC of 0.87, the naïve Bayes algorithm showed the best performance parameters. As described above, the model did not improve significantly (>5) by adding another item. Under the inclusion of all initial 107 possible predictor variables, the AUC yielded 0.89 as compared to the AUC of 0.87 under the inclusion of 6 variables. This very small delta between the two AUC demonstrates that the model is mainly dominated by the 6 predictor variables. The absolute and relative distribution of the 6 most predictive variables identified during nested resampling, which were used for the model building, are shown in Table 3.

The quality of the final naïve Bayes model on the validation subset is provided in Table 4.

While the balanced accuracy of 71.2% and the AUC of 0.81 were a little lower than the results of the initial training model, they were still indicative. With a sensitivity of 61.5% and a specificity of 80.9%, NOP with a history of suicidality were identified correctly in two-thirds of the cases, while four-fifths of cases were identified correctly as being OP with a history of suicidality.

### 3.3. Ranking of Predictor Variables

Figure 2 shows the effect on the output variable (NOP/OP) by varying each predictor variable at a time, keeping all the other predictor variables at their initial values. The x-axis represents the relative variable importance, and the y-axis each variable (the wider the bar, the more impact the variable has on the model and the outcome). Consequently, the predictor variables are ranked from the most influential to the least influential within the model.

The two variables most indicative of being a non-offender patient in the model referred to as pharmacotherapy were “antidepressant during referenced hospitalisation” and “regular intake of antipsychotic medication”. Other influential items that set apart NOP from OP were having had “any outpatient psychiatric treatment(s) in the past”, having a “global cognitive deficit” as well as higher scores on the adapted PANSS items “anxiety” and “lack of spontaneity and flow of conversation”.

## 4. Discussion

The aim of this study was to determine the factors that distinguish between patients with SSD and a history of suicidality who had committed a criminal offence from those who did not. Regarding demographic and psychiatric characteristics, we found that OP were more often from native countries other than Switzerland, and, as expected, showed a significantly higher rate of endangerment of others both in the past and during the referenced hospitalisation. Even though it did not quite reach a satisfying level of significance, NOP had more frequently attempted suicide in the past. Apart from these characteristics, both groups were rather similar, thus allowing optimal comparability.

As there is little research on the matter, thus complicating the generation of null hypotheses, we used and explorative approach, applying ML algorithms to a large database consisting of 399 patients matched for age and gender. In doing so, we were able to create an appropriate model: With a balanced accuracy of 71% and an AUC of 0.81, the model based on the Naïve Bayes algorithm as able to correctly identify non-offenders in two-thirds of the cases. Variables related mostly to integration into the therapeutic support network, pharmacotherapy, and cognition. Results painted a picture of the NOP with a history of suicidality as being more compliant regarding (pharmaco-)therapy—which was more likely to include antidepressants, but more subjected to cognitive deficits and anxiety. They were more likely to have a prescription for *antidepressants during the referenced hospitalisation*. This observation is not only found when comparing merely the patients with a suicidal history: a higher rate of antidepressant pharmacotherapy in NOP with SSD than OP with SSD has been observed in general [33]. While an adjunctive antidepressant can be beneficial to patients with SSD, especially regarding negative symptoms, it seems to be a less common prescription in OP, possibly due to fear of re-exacerbation of positive psychotic symptoms and an increased risk of side effects [34,35]. This could also prevent clinicians from targeting suicidality with antidepressants. They may rather rely on antipsychotics and mood stabilisers in suicidal offender patients, e. g. clozapine, which is especially recommended for forensic psychiatric populations due to evidence for its efficacy against violent behaviour but has also shown to be effective regarding suicidality [36,37]. As research on prescription practices in offender populations with SSD are scarce, the possible reasons that explain why antidepressant medication is the most influential variable in the model remain hypothetical and cannot be confirmed without data on the clinicians’ decision process in pharmacotherapeutic treatment.

Compared to OP, NOP were also more likely to have been involved in psychiatric treatment prior to the referenced hospitalisation in the form of outpatient treatment and to show a regular intake of their prescribed antipsychotic medication. In the literature, a negative attitude towards therapy or a lack of compliance has been described as a risk factor for the development of suicidality in general, but not for forensic psychiatric patients with SSD [20,38,39]. The question remains as to why this factor emerges as a distinguishing feature between OP and NOP even in a sample exclusively comprising patients who had shown suicidality in the past. Previous research has described an insufficient integration into the mental health care system in offenders with SSD: while there is high utilisation of inpatient treatments, regular outpatient therapies and regular intake of prescribed pharmacotherapeutic substances are observed less frequently [40,41]. It has been hypothesised, that due to a greater risk profile, inpatient care becomes more likely in individuals with violent behaviour [42]. This could largely refer to compulsory admissions, which are indicated and legally permissible in Switzerland in cases of acute danger to the self, but also to others. In summary, the current findings indicate that—while lack of compliance regarding (pharmaco-)therapeutic treatment is a known risk factor of suicidality—the risk factor domain of insufficient treatment is particularly salient in a population that is clustered to show inadequate treatment adherence.

NOP also showed a higher prevalence of global cognitive deficits and scored higher on the PANSS-item “lack of spontaneity and flow of speech”, which seems likely to be interlinked. This was surprising, as previous research comparing OP and NOP with SSD found the former to have worse overall cognitive functioning [43]. However, it has been hypothesised that institutionalised OP may participate in cognitive tests with greater engagement to demonstrate motivation, resulting in better scores when compared to their controls [44]. There is also a possible selection bias to be considered: Data on the NOP group were extracted from patients admitted to a rehabilitative psychiatric institution, focusing on reintegration into social and, if appropriate, working life. Due to this treatment focus, physicians are likely to refer patients who primarily have limited daily living skills rather than impairments in other symptom domains, e. g. positive productive symptoms, as focus of their therapeutic goals. In contrast, admittance to the institution from which the OP data stem is not determined by symptomatology of the individual patient, but by court mandate. The OP sample may therefore not be as subjected to pre-selection of patients corresponding to the NOP sample.

Lastly, NOP were characterised by higher levels of anxiety as measured by an adapted PANSS than their OP counterparts. Anxiety has been identified as a mediator for suicidality in both offender populations and SSD populations [45,46,47,48]. At the same time, anxiety can be difficult to detect in patients with SSD due to obscuring positive psychotic symptoms and impairment regarding the ability to verbalise and express emotions [47]. This could even more be the case in offender populations, as they may tend to externalise anxiety through antisocial and impulsive behaviours as an outward expression of emotional distress, as described by Carragher et al. [49]. Another possible explanation is a potentially reduced openness on the part of the patient in the coercive context of the forensic psychiatric setting: The special role of the forensic psychiatrist as a “dual mandate holder” who not only treats but also controls, orders, and monitors leads to a certain vertical relationship since the patient is not able to simply terminate therapy without consequences [50]. This may consequently have an influence on trust in the treating clinician. Especially in the initial treatment period, a longer phase of therapeutic relationship building is usually necessary to allow the patient to verbalise and show vulnerability. However, as Höfer et al. could demonstrate, the quality of the therapeutic relationship is not associated with the patient’s legal status, but with the severity of hostility as a symptom, thus showing that a coercive setting does not obligatorily lead to an impaired therapeutic basis [51].

Interestingly, variables linked to aggression, although more prevalent in the offender population, were not amongst the most influential variables in the model and were dominated by the items discussed above. Even though over half of the patients in both groups had shown some kind of endangerment of others in the past and OP had a significantly higher expression of aggression towards others both before and during their referenced hospitalisation, the item did not emerge as highly powerful in distinguishing the two groups from another. It can however not be ruled out, that variables regarding aggression could have more impact if a more precise distinction is made between the different manifestations of aggression. When taking previous literature on offenders into account, this seems likely, as the severity of the offence committed has been associated with the risk of suicide [52,53]. The current finding also indicates that aggression against others and suicidality are indeed independent phenomena. While such a linkage has been described in previous literature for adults and adolescents, other characteristics present in patients with a pattern of aggressive behaviour may be the driving forces behind suicidal developments [8,9,54].

Another striking finding was that, while individual symptom domains such as anxiety and cognitive deficits were of great influence in the model, overall clinical impairment as measured by the total PANSS score was not. The few studies directly comparing offender and non-offender patients with SSD reported the former to be more clinically impaired, which seems logical, as court-mandated treatment is mostly intended for highly severe courses of the disorder [20,43,55]. A potential explanation for this contradictive result could be in the selection of our comparison group: As described in the methodology section, we chose patients within a rehabilitative psychiatric institution due to their similar rate of chronically ill patients. It could be that the overall burden of disease had less influence on the model than other items because of similarly severe treatment courses in both study and comparison groups.

When looking into the limitations of this study, the most obvious is retrospective design. While we aimed at ensuring sufficient quality of data by using a structured data extraction protocol, data quality comparable to a prospectively standardised study cannot be assumed. This is especially the case for items that may be subject to high interrater reliability, such as “social isolation”. Such items may have already been scored differently by differing clinicians during documentation, thus resulting in skewed results. For robust variables, such as the number of previous hospitalisations, it is likely that this effect is less pronounced. Additionally, as the data extraction was performed by two psychiatrists rather than one, it cannot be ruled out that biases were introduced through this procedure. When looking at the selection of our comparison group, it should be noted that some of the NOP had also shown aggressive incidents in the past. Nevertheless, those patients were not grouped into the OP sample as their aggressive incidents had remained without the involvement of the judicial system in a sense of court-mandated treatment.

Another limitation was our broad definition of suicidality, including suicidal thoughts and ideations, plans, and attempts. The rationale behind this decision was the intention to cover suicidality as a whole entity. However, it cannot be ruled out that results may present themselves differently when differentiating more between the different manifestations of suicidality. In general, while our sample can be considered rather large for forensic psychiatry as a niche specialty, it has to be acknowledged that the total case number of 399 patients, each group collected from a single institution, can merely serve exploratory purposes. Further application and validation of the model to a larger population are, therefore, recommended, preferably in a multicentric approach to eliminate possible bias through characteristics of the institution. Lastly, as expected in offender populations with matching controls, our sample was predominantly male, thus limiting the applicability of our results to women in forensic psychiatric institutions. Replication of our findings in a larger sample, therefore, seems sensible, preferably in a prospective approach and, if possible, under the inclusion of more female patients.

In summation, the present findings enhance our understanding of suicidality in offender and non-offender patients with SSD and their differing characteristics. Using ML, we identified the 6 factors most distinguishing between the two groups, as well as their complex interplay out of a large dataset with 107 possibly predictive different parameters. The results suggest that even though both OP and NOP have an elevated risk for suicidality, different risk factors could have differing weightings depending on the population. This should raise awareness amongst clinicians when evaluating suicidality: Not all risk factors seem to have the same impact in different patient populations, and certain interventions may therefore be more indicated and profitable for one group than the other. For instance, it seems sensible to further address anxiety in OP with a history of suicidality. In general, the authors propose further research on suicidality in these populations, ideally in prospective trials with a large population as to allow subgroup testing as well as generalizability.

## Figures and Tables

**Figure 1 brainsci-13-00097-f001:**
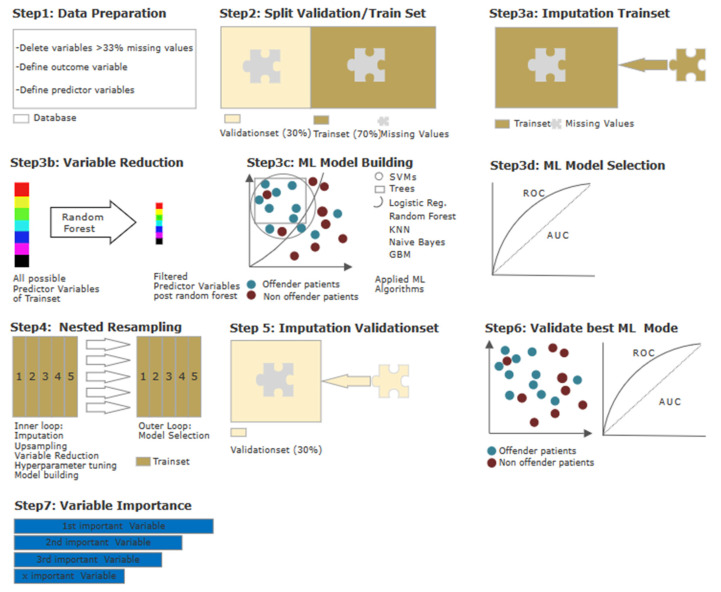
Statistical procedures using machine learning (ML). Step 1—Data preparation: Conversion of categorial variables to binary code. Outcome variable OP/NOP and predictor variables were defined. Omission of variables with missing values of 30% or more Step 2—Data splitting: 70% training dataset and 30% validation dataset. Step 3a–d—Model building and selection (a): Imputation by mean; (b) variable reduction via random forest; (c) model building using ML algorithms (d) selection of most suitable ML algorithm via ROC parameters. Step 4—Model building and testing on training subset: Step 5—Imputation by mean on validation set. Step 6—Model building and testing on validation data: Application of the most suitable model identified in Step 3c on imputed validation dataset, evaluation via ROC parameters. Step 7—Test for multicollinearity and ranking of variables.

**Figure 2 brainsci-13-00097-f002:**
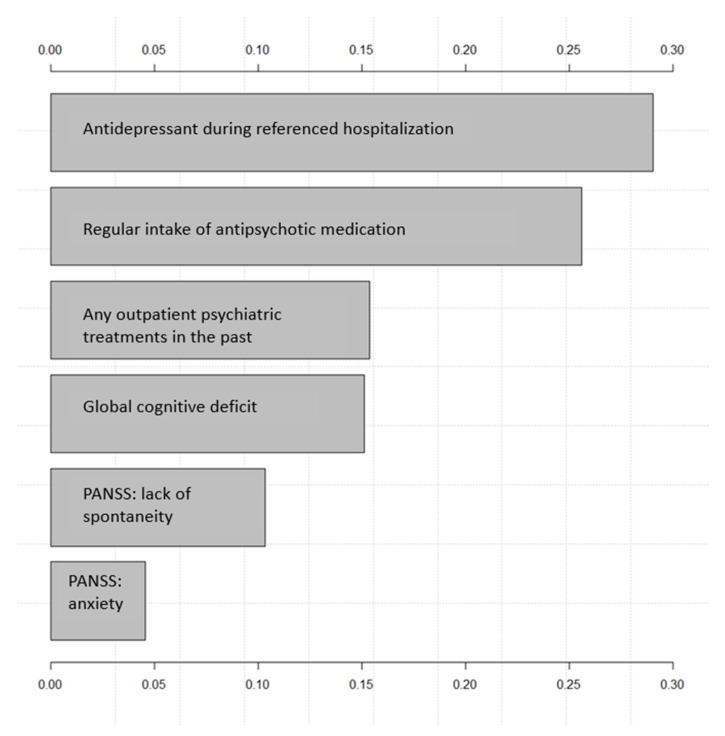
Importance of variables by naïve bayes. PANSS refers to adapted PANSS as described in the methodology section.

**Table 1 brainsci-13-00097-t001:** Demographic characteristics of the study population.

Characteristics	Total n/N (%)	Non-Offender Patients n/N (%)	Offender Patients n/N (%)	*p*-Value
Male sex	363/399 (91)	152/167 (91)	211/232 (90.9)	1
Age at admission (mean, SD)	34.4 (10.7)	35.7 (11.9)	33.5 (9.6)	0.189
Native Country Switzerland	210/398 (52.8)	105/166 (63.3)	105/232 (45.3)	0.001 *
Single ^1^	303/399 (75.9)	123/166 (74.1)	180/232 (77.6)	0.335
Diagnosis: Schizophrenia	308/399 (77.2)	129/166 (77.7)	179/229 (79)	0.904
Suicide attempt in past	208/375 (55.5)	92/149 (61.7)	116/226 (51.3)	0.056
Suicide attempt during referenced hospitalisation	12/388 (3)	2/163 (1.2)	10/225 (4.4)	0.081
Endangerment of others in past	260/381 (65.2)	90/152 (59.2)	170/229 (74.2)	0.002 *
Endangerment of others during referenced hospitalisation	105/387 (26.3)	33/165 (20)	72/222 (32.4)	0.008 *

SD = standard deviation; N = total study population; n = subgroup with corresponding characteristic. ^1^ at the time of the investigated offence resp. at the time of admission to referenced hospitalisation. * indicating statistical significance; level of significance *p* < 0.05.

**Table 2 brainsci-13-00097-t002:** Machine learning models and performance in nested cross-validation.

Statistical Procedure	Balanced Accuracy (%)	AUC	Sensitivity (%)	Specificity (%)	PPV (%)	NPV (%)
Logistic Regression	62.50	75.70	79.30	45.50	50.40	76.90
Tree	67.50	75.60	85.20	49.80	54.40	82.80
Random Forest	66.2	75.7	84	48.3	52.9	82.2
Gradient Boosting	76.9	0.85	65.1	88.7	78.6	79.4
KNN	64.1	76.5	81.9	46.3	51.3	80.2
SVM	73.7	0.85	54.2	93.2	84.4	74.7
**Naïve Bayes**	**76.6**	**0.87**	**63.6**	**89.7**	**81.1**	**77.7**

AUC = area under the curve (level of discrimination); PPV = positive predictive value; NPV = negative predictive value; KNN = k-nearest neighbours; SVM = support vector machines. Bold font indicates the algorithm with the best performance measures. Bold font indicates the algorithm with best performance measures.

**Table 3 brainsci-13-00097-t003:** Absolut and relative distribution of predictor variables most dominant in the model.

Variable Code *	Variable Description	NOP n/N (%)	OP n/N (%)
PH18a	Any outpatient psychiatric treatment(s) in the past	**134/153 (87.6)**	130/218 (59.6)
PH23p	Regular intake of antipsychotic medication	**80/143 (55.9)**	17/150 (11.3)
N2	Global cognitive deficit	**133/152 (87.5)**	148/230 (64.3)
R9l	Antidepressant during current hospitalisation	**63/167 (37.7)**	23/195 (11.8)
PA13	PANSS—adapted scale at admission: Lack of spontaneity and flow of conversation		
	symptom absent	77/166 (46.4)	**144/224 (65.2)**
	symptom discreetly	**48/166 (28.9)**	33/224 (14.7)
	symptom substantially	**41/166 (24.7)**	47/224 (21)
PA47	PANSS—adopted scale at admission: Anxiety		
	symptom absent	79/52.7 (52.7)	**170/224 (75.9)**
	symptom discreetly	**57/150 (38)**	38/224 (17)
	symptom substantially	**14/150 (9.3)**	16/224 (7.1)

PANSS = positive and negative syndrome scale. Bold font indicates the group in which the respective item is expressed with a higher rate. * Variable coding is used for better traceability and comparability (please refer to the Appendix A and Appendix B for a detailed definition of each variable). Coding was not performed specifically for this study but within the framework of our larger research project.

**Table 4 brainsci-13-00097-t004:** Performance measures of naïve Bayes model on the validation set.

Performance Measures	% (95% CI)
Balanced Accuracy	71.2 (62.4–78.3)
AUC	0.81 (0.73–0.89)
Sensitivity	61.5 (47.0–74.4)
Specificity	80.9 (69.2–89.0)
PPV	71.1 (55.5–83.2)
NPV	73.3 (61.7–82.6)

AUC = area under the curve (level of discrimination); PPV = positive predictive value; NPV = negative predictive value.

## Data Availability

The datasets generated and analysed during the current study as well as a detailed list are of all our variables (including definitions and references) are available from the corresponding author on request.

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
