# Peer review of "Suicidal Offenders and Non-Offenders with Schizophrenia Spectrum Disorders: A Retrospective Evaluation of Distinguishing Factors Using Machine Learning"

_brainsci, 2023, doi:10.3390/brainsci13010097_

Round 1
Reviewer 1 Report
In this study, inpatients with schizophrenia and related disorders, all of which had a history of suicidal ideation or behavior, were compared according to whether they had a history of criminal offences. Using machine learning algorithms the authors found mainly clinical factors to discriminate between the two groups. Some of these are not plausible at first sight (e.g, one would expect that for forensic SSD inpatients it would be mandatory to take antipsychotic medication - or did they more often receive injectable APs?).
While the topic in general is of interest, there are a number of issues which make an in-depth revision necessary.
· The Introduction reports on suicide risk factors in SSD patients and the reader feels guided towards a manuscript on this issue. However, then its focus is a completely different one. Therefore, the Introduction needs a thorough revision.
· Did the authors consider to study the entire 740 patients sample and include suicidal items as dependent factors? This approach would potentially result in more persuasive findings. Please clarify why suicidality was an inclusion criterion.
· In addition, the title is at least misleading. It were not differences in suicidality that were studied.
· In the title, the retrospective nature of the study should be stated. Probably because of this background no ethics committee decision was required? Or is there one?
· Were the files of both groups rated by the same author(s)? Or one rater for OP and another for NOP patients?
· I understand that data were assessed clinically. How can the authors be sure that a forensic/violent history can clearly be ruled out in the NOP group?
· Is Table 1 really necessary?
· It is not clear what "relevant" in Table 4 does mean (significance?). SD is explained in the table's legend but does not appear here.
· It is not explained how the variable codes in Table 4 and the appendix are categorized (PH, N, R,...). Was it done specifically for the study?
· Abstract: Please state OP and NOP numbers separately.
· Please check for misspellings (e.g., line 11 (?); endangerment in Table 2; line 215 (in?); line 239 (and); ...).
Author Response
“While the topic in general is of interest, there are a number of issues which make an in-
depth revision necessary.”
Thank you for your careful and highly appreciated review. Please find below our answers to each comment.
“The Introduction reports on suicide risk factors in SSD patients and the reader feels
guided towards a manuscript on this issue. However, then its focus is a completely different
one. Therefore, the Introduction needs a thorough revision.”
Through the introduction, we tried to outline that while suicidality occurs frequently in SSD patients, certain subgroups, like offender patients with SSD, are at an increased risk and may be influenced by other mediators than non-offender patients with SSD. We have adapted the introduction to make this more clear to the reader in the hopes to have met your expectations.
“Did the authors consider to study the entire 740 patients sample and include suicidal
items as dependent factors? This approach would potentially result in more persuasive
findings. Please clarify why suicidality was an inclusion criterion.”
We chose suicidality as inclusion criterion rather than as independent variable, as our independent variable was offender patient (yes/no) – considering that our aim was not to identify group differences between OP and NOP in general, but specifically group differences between suicidal OP and suicidal NOP (Machine Learning classifiers usually support a single target variable/independent variable). We thought this design would be most suitable for the special issue’s topic.
“In addition, the title is at least misleading. It were not differences in suicidality that
were studied.”
We understand how the title is misleading. We suggest the following title instead: “Suicidal offenders and non-offenders with schizophrenia spectrum disorders: A Retrospective Evaluation of Distinguishing Factors using Machine Learning”.
“In the title, the retrospective nature of the study should be stated. Probably because of
this background no ethics committee decision was required? Or is there one?”
We agree, please see the answer above regarding the title.
In Switzerland, both prospective and retrospective trials need to be approved by the ethics committee. The study was reviewed and approved by the Ethics Committee Zurich and information on the reference number is stated in the institutional review board statement at the bottom of the manuscript. We have now also included this information at the end of the introduction.
"Were the files of both groups rated by the same author(s)? Or one rater for OP and another for NOP patients"
We have clarified the data extraction procedure, which was performed by two raters: “Data extraction was performed by two experienced psychiatrists according to an adapted rating protocol based on a set of criteria originally described by Seifert & Nedopil.”
"I understand that data were assessed clinically. How can the authors be sure that a
forensic/violent history can clearly be ruled out in the NOP group?"
The NOP sample was also screened for their criminal history and history of violent behavior. Some patients had shown aggressive incidents without involvement of the judicial system in a sense of court mandated treatment in the past. If you consider numbers on this aspect valuable to the reader, we will gladly provide it, e.g. in a tabular overview over previous aggressive incidents in both groups as well as previous detentions/penitentiary measures.
“Is Table 1 really necessary?”
We considered the table to be useful in order to show why we chose to explore the dependent variables of our study (first column), which were selected based on previous literature (second column). A tabular overview seemed to be the most practicable. If you feel that it should be omitted or is more suitable as appendix, we will gladly leave it out respectively leave it in the appendix.
“It is not clear what "relevant" in Table 4 does mean (significance?). SD is explained in
the table's legend but does not appear here.”
We have clarified the term as follows: “Absolut and relative distribution of predictor variables most dominant in the model”.
The SD is indeed nowhere in the table, thank you for bringing this to our attention. We have corrected the mistake in the footer.
“It is not explained how the variable codes in Table 4 and the appendix are categorized
(PH, N, R,...). Was it done specifically for the study?”
Variable coding is used for better traceability and comparability (please refer to the appendix for a detailed definition of each variable). Coding was not performed specifically for this study but within the framework of our larger research project. We have provided this information in the footer of the table. However, we still opted to leave in the codes in the table as it allows readers to find the detailed variable definition in the appendix as well as in the coding sheet of the larger research project (of which this study was part), which is available as stated data availability statement.
“Abstract: Please state OP and NOP numbers separately.”
We have now changed the abstract with separate numbers for OP and NOP instead of the total of the overall population.
“Please check for misspellings (e.g., line 11 (?); endangerment in Table 2; line 215
(in?); line 239 (and); ...)”
Thank you for bringing these to our attention. We have corrected the typos you have pointed out as well as double-checked the manuscript for further necessary corrections of misspellings and the like.
Reviewer 2 Report
Suicidality in offenders and non-offenders with schizophrenia spectrum disorders: What are the differences?
Thank you for the opportunity to review this interesting article. The article is well written and constitutes an important contribution to the field. Nevertheless, some changes would improve the overall quality of the manuscript:
1. Authors should provide more information on why this particular population can be considered dual-vulnerable. What intersectional factors can be detrimental to their well-being?
2. More information regarding the positive and negative syndrome scale (PANSS) must be provided, namely, its reliability scores.
3. Despite the fact that machine learning procedures have been described previously elsewhere, readers would benefit from a brief description here.
4. Implications of these results in terms of mental health and social policies should be addressed.
Best wishes.
Author Response
“Thank you for the opportunity to review this interesting article. The article is well written and constitutes an important contribution to the field. Nevertheless, some changes would improve the overall quality of the manuscript:”
Thank you for your diligent comments and suggestions to improve our manuscript. Please find our answers below.
- “Authors should provide more information on why this particular population can be considered dual-vulnerable. What intersectional factors can be detrimental to their well-being?”
We consider this population to be dual-vulnerable due to a) the coercive treatment context of a court mandated therapy and b) the underlying SSD with oftentimes severe impairments in many domains. We have added this to the corresponding section (see line 55 -56).
- “More information regarding the positive and negative syndrome scale (PANSS) must be provided, namely, its reliability scores.”
We have added another paragraph briefly describing the PANSS and providing reliability scores as well as interrater correlation, see line 108 ff.
- Despite the fact that machine learning procedures have been described previously elsewhere, readers would benefit from a brief description here.
We have adapted the methodology section in the hopes to have met your expectations. If there still are certain aspects that may be unclear to the readers, we will of course gladly expand the section further.
- Implications of these results in terms of mental health and social policies should be addressed.
We have elaborated on clinical implications in our discussion, please see line 439 ff.
Reviewer 3 Report
Thank you for the opportunity to review this interesting manuscript. The study is relevant given that there are hardly any studies that analyze suicidality in schizophrenia spectrum disorders comparing offenders and non-offenders.
The authors carry out an adequate theoretical review. In addition, they use a novel method to test their hypotheses. I only consider that some issues could be clarified to improve the understanding of the manuscript.
1. At the end of the first paragraph of the introduction, specify which is the association that appears between variables: “reduced risk of suicide and…”.
2. There is a typographical error in the second paragraph of the introduction. There should be an equal sign after the OR.
3. In the materials and methods section, perhaps it could be clarified how suicidality was measured before the participants were hospitalized. How were the participants contacted prior to hospitalization? or was that information assessed retrospectively?
4. I don't quite understand which were the predictor variables used. The authors cite 32 variables in Table 1 but later comment that they included 107 possible predictor variables (Appendix).
5. In Table 1, the variables that measure PANSS p1-p7, PANSS N1-N7, PANSS G1-G16 and PANSS total should appear at the foot of the table.
6. Finally, the authors present in the results a model with the 6 most predictive variables, but they do not explain how the cut-off point was to keep only 6 of the 107.
7. In table 4, I think there is a typographical error in PA13 "adapted" appears and in PA47 "adopted” appears.
8. In the discussion section, the authors mention that the 399 participants were “matched by age and gender”. This information should appear in procedure. In addition, if the 399 participants who met the suicidality criteria were selected from the initial sample, How did the authors match the groups of offenders and non-offenders based on age and gender?
Please clarify the issues indicated.
Best regards,
Author Response
“Thank you for the opportunity to review this interesting manuscript. The study is relevant given that there are hardly any studies that analyze suicidality in schizophrenia spectrum disorders comparing offenders and non-offenders. The authors carry out an adequate theoretical review. In addition, they use a novel method to test their hypotheses. I only consider that some issues could be clarified to improve the understanding of the manuscript.”
Thank you for your careful and highly appreciated review. Please find below our answers to each comment.
“1. At the end of the first paragraph of the introduction, specify which is the association that appears between variables: “reduced risk of suicide and…”.”
We have now specified the findings suggesting a reduced risk of suicide and hallucinations, see line 39ff.
“2. There is a typographical error in the second paragraph of the introduction. There should be an equal sign after the OR.”
Thank you for pointing out this typo. We have corrected it accordingly.
"3. In the materials and methods section, perhaps it could be clarified how suicidality was measured before the participants were hospitalized. How were the participants contacted prior to hospitalization? or was that information assessed retrospectively?"
Yes, this was also assessed retrospectively through the extensive case files. We have added this information in the methodology section (see line 117 ff).
"4. I don't quite understand which were the predictor variables used. The authors cite 32 variables in Table 1 but later comment that they included 107 possible predictor variables (Appendix)."
The variables in table 1 partially comprise several items (e.g. PANSS: P1-P7 contains 14 individual variables). For the purpose the readability and overview, we decided to summarize variables in table 1, and elaborate on each of the variables in the appendix.
“5. In Table 1, the variables that measure PANSS p1-p7, PANSS N1-N7, PANSS G1-G16 and PANSS total should appear at the foot of the table.”
We have added the items as well as the possible range of the total PANSS in the footer.
“6. Finally, the authors present in the results a model with the 6 most predictive variables, but they do not explain how the cut-off point was to keep only 6 of the 107.”
We had briefly mentioned it in the methodology section (see line 189-190), but absolutely agree that this aspect needs to be explained in the results section as well to provide readers with a better understanding of the rationale behind this decision. We have added this now in section 3.2.
“7. In table 4, I think there is a typographical error in PA13 "adapted" appears and in PA47 "adopted” appears.”
Thank you for pointing out this typo. We have corrected it in the table.
"8. In the discussion section, the authors mention that the 399 participants were “matched by age and gender”. This information should appear in procedure. In addition, if the 399 participants who met the suicidality criteria were selected from the initial sample, How did the authors match the groups of offenders and non-offenders based on age and gender?"
We have included information regarding matching in the methodology section. Matching was performed in the initial sample before omitting cases that did not meet the inclusion criterion of suicidality. As we wanted to match both groups by gender, no further female patients were included in the NOP sample after having reached the number of female patients in the OP group.
Reviewer 4 Report
Authors show a study model of the characteristics of patients at risk of suicide among psychiatric patients, in the cases of offenders and non-offenders, using Machine Learning models.
[Line 37]: please remove the capital letter. "...hallucinations are mixed: While some authors..." should be "...hallucinations are mixed: while some authors...".
[Line 241] same typo: "...model: With a ..."
Please look if there are some repetitions. for example, in line 230 you repeat your aims, in line 241 you repeat what you said in line 212.
However, the bibliography needs to be expanded with additional articles pertaining to various suicide dynamics. In a discussion of psychiatric Disorder's suicides, I expect to read a minimal hint of the modalities. I propose these two articles:
- A special case of suicide enacted through the ancient Japanese ritual of Jigai. The American journal of forensic medicine and pathology, 35(1), 8-10. https://doi.org/10.1097/PAF.0000000000000070
- Determined to Die! Ability to Act Following Multiple Self-inflicted Gunshot Wounds to the Head. The Cook County Office of Medical Examiner Experience (2005-2012) and Review of Literature. Journal of forensic sciences vol. 60,5 (2015): 1373-9. doi:10.1111/1556-4029.12780
The manuscript is clear and well-structured manner. Congratulations. Please provide this few corrections.
Author Response
"Authors show a study model of the characteristics of patients at risk of suicide among psychiatric patients, in the cases of offenders and non-offenders, using Machine Learning models."
Thank you for your diligent revisions. Please find our according changes and answers below.
“[Line 37]: please remove the capital letter. "...hallucinations are mixed: While some authors..." should be "...hallucinations are mixed: while some authors...". [Line 241] same typo: "...model: With a ..."”
Thank you for bringing these to our attention. We have corrected both typos.
“Please look if there are some repetitions. for example, in line 230 you repeat your aims, in line 241 you repeat what you said in line 212.”
We have now double-checked and rearranged some passages in order to avoid repetitions.
“However, the bibliography needs to be expanded with additional articles pertaining to various suicide dynamics. In a discussion of psychiatric Disorder's suicides, I expect to read a minimal hint of the modalities. I propose these two articles:
- A special case of suicide enacted through the ancient Japanese ritual of Jigai. The American journal of forensic medicine and pathology, 35(1), 8-10. https://doi.org/10.1097/PAF.0000000000000070
- Determined to Die! Ability to Act Following Multiple Self-inflicted Gunshot Wounds to the Head. The Cook County Office of Medical Examiner Experience (2005-2012) and Review of Literature. Journal of forensic sciences vol. 60,5 (2015): 1373-9. doi:10.1111/1556-4029.12780”
We highly appreciate your input and the article propositions, which we have gladly incorporated in our discussion.
Round 2
Reviewer 1 Report
· The manuscript clearly has improved but still the Introduction is too much focused on general suicide risk factors in SSD patients while it should be on the differences between the two populations.
· The fact that the groups were rated by different researchers should be stated as a limitation as should be the potential that NO patients also had an offence-like history.
· To transfer table 1 into the appendix is a good compromise.
· Table 4: "most dominant" is still not clear. Is there a cut-off value or something scientifically graspable?
Author Response
Thank you for taking the time to clarify some aspects of the revision. Please find the answers to each comment below.
“The manuscript clearly has improved but still the Introduction is too much focused on general suicide risk factors in SSD patients while it should be on the differences between the two populations.”
We have added a paragraph describing why results from general psychiatric patients may not be directly applicable to OP, and have elaborated on known systematic differences between the two comparison groups (see ll. 66ff).
“The fact that the groups were rated by different researchers should be stated as a limitation as should be the potential that NO patients also had an offence-like history.”
We have now added both aspects and the possible bias as their consequence as limitations in the discussion section.
“Also, as the data extraction was performed by two psychiatrists rather than one, it can’t be ruled out that biases were introduced through this procedure. When looking at the selection of our comparison group, it should be noted that some of the NOP had also shown aggressive incidents in the past. Nevertheless, those patients were not grouped into the OP sample as their aggressive incidents had remained without involvement of the judicial system in a sense of court mandated treatment.”
“To transfer table 1 into the appendix is a good compromise.”
We are happy that you feel content about our suggestion and have moved table 1 into the appendix, and have of course adjusted numbering of the other remaining tables in the main text.
“Table 4: "most dominant" is still not clear. Is there a cut-off value or something scientifically graspable?”
We understand this needs clarification. There is in fact no cut-off value in explorative analysis. Instead, the model is built in accordance with the highest AUC, in in broad terms the model's ability to predict classes correctly. In this case, the AUC did not improve by more than 5% (from 0.87 to >92) through adding another item. In fact, if all 107 variables were included in the model, the AUC yielded 89%, meaning that even with inclusion of all variables in the model, the probability to correctly predict class increase only by 0.02 – we have now elaborated on that in the results section (see ll.258-264). This shows the dominance of the 6 variables in table 3 (formerly table 4) in the model.
In addition, we have adapted section 3.3. and elaborated on the one-sided tornado graph in the hopes to provide readers with a better understanding of how to interpret the figure.